# Genome-Wide Identification and Analysis of the *NPR1*-Like Gene Family in Bread Wheat and Its Relatives

**DOI:** 10.3390/ijms20235974

**Published:** 2019-11-27

**Authors:** Xian Liu, Zhiguo Liu, Xinhui Niu, Qian Xu, Long Yang

**Affiliations:** 1State Key Laboratory of Crop Biology, College of Agronomy, Shandong Agricultural University, Taian 271018, China; liuxian123abc@163.com (X.L.); niuxh1227@163.com (X.N.); 2Agricultural Big-Data Research Center, College of Plant Protection, Shandong Agricultural University, Taian 271018, China

**Keywords:** NPR1, wheat, phylogenetic analysis, *Fusarium graminearum*, expression profile, biotic stress

## Abstract

NONEXPRESSOR OF PATHOGENESIS-RELATED GENES 1 (NPR1), and its paralogues NPR3 and NPR4, are bona fide salicylic acid (SA) receptors and play critical regulatory roles in plant immunity. However, comprehensive identification and analysis of the *NPR1*-like gene family had not been conducted so far in bread wheat and its relatives. Here, a total of 17 *NPR* genes in *Triticum aestivum*, five *NPR* genes in *Triticum urartu*, 12 *NPR* genes in *Triticum dicoccoides*, and six *NPR* genes in *Aegilops tauschii* were identified using bioinformatics approaches. Protein properties of these putative *NPR1*-like genes were also described. Phylogenetic analysis showed that the 40 NPR1-like proteins, together with 40 NPR1-related proteins from other plant species, were clustered into three major clades. The *TaNPR1*-like genes belonging to the same *Arabidopsis* subfamilies shared similar exon-intron patterns and protein domain compositions, as well as conserved motifs and amino acid residues. The cis-regulatory elements related to SA were identified in the promoter regions of *TaNPR1*-like genes. The *TaNPR1*-like genes were intensively mapped on the chromosomes of homoeologous groups 3, 4, and 5, except *TaNPR2-D*. Chromosomal distribution and collinearity analysis of *NPR1*-like genes among bread wheat and its relatives revealed that the evolution of this gene family was more conservative following formation of hexaploid wheat. Transcriptome data analysis indicated that *TaNPR1*-like genes exhibited tissue/organ-specific expression patterns and some members were induced under biotic stress. These findings lay the foundation for further functional characterization of NPR1-like proteins in bread wheat and its relatives.

## 1. Introduction

Bread wheat (*Triticum aestivum* L.), also known as common wheat, is a major grain crop with very high economic value globally. This allohexaploid bread wheat (*T. aestivum*, 2*n* = 6x = 42) is comprised of three closely homoeologous chromosome groups (AABBDD), originating from a series of natural hybridization events [1,2]. Firstly, diploid *Triticum urartu* (the A-genome donor) hybridized with an unknown diploid grass (considered probably as *Aegilops speltoides*, the B-genome donor) to produce tetraploid emmer wheat (*Triticum dicoccoides*, AABB). Subsequently, emmer wheat hybridized with diploid goat grass (*Aegilops tauschii*, the D-genome donor) to form hexaploid wheat around 8000 years ago. In field conditions, bread wheat is challenged by various fungal pathogens, such as *Fusarium graminearum* (*Fusarium* head blight, *FHB*), *Puccinia striiformis* (stripe rust), and *Blumeria graminis* (powdery mildew), which cause huge losses in yield and quality [3]. The salicylic acid (SA) signaling pathway is required for plant immunity against biotrophic and hemi-biotrophic pathogens [4,5]. NONEXPRESSOR OF PATHOGENESIS-RELATED GENES 1 (NPR1), and its paralogues NPR3 and NPR4, are *bona fide* SA receptors that act as master regulators in SA-mediated local and systemic immunity (also named systemic acquired resistance, SAR) [6,7,8,9]. Therefore, identification and analysis of these essential components involved in SA-dependent defense responses is very important to understand the immune mechanism in bread wheat and its relatives.

Plants have evolved a highly sophisticated and effective innate immune system to combat pathogens, including bacteria, fungi, viruses, and oomycetes [10,11,12]. Facing the pathogen challenges, the first line of defense initiates on the plant cell surface, called pathogen-associated molecular pattern (PAMP)-triggered immunity (PTI) [13,14]. Pattern-recognition receptors (PRRs) at the plasma membrane activate PTI responses by detection of PAMPs, such as fungal chitin, bacterial flagellin (flg22), and lipopolysaccharides (LPS) [15,16,17]. However, many microbial pathogens often secrete effectors in the infection process, which can depress PTI and lead to effector-triggered susceptibility (ETS) [18,19,20]. During evolution, plants have developed the second layer of local induced resistance, termed effector-triggered immunity (ETI) [21,22]. Plant intracellular sensors encoded by resistance (*R*) genes elicit ETI responses by recognizing these attacker-specific effectors. The *R* gene-mediated defenses confer strong resistance and effectively restrict the growth of pathogens via programmed cell death (PCD), designated as hypersensitive response (HR). This local immune response leads to biosynthesis and accumulation of plant defense hormone SA both at infection sites and in distal uninfected tissues, and then deployment of systemic acquired resistance (SAR) after HR in the whole plant. SAR confers a broad-spectrum, long-lasting, and systemic resistance to secondary infections, that is characterized by expression of many anti-microbial pathogenesis-related (*PR*) genes throughout plant’s tissues [23,24,25].

In the mid-1990s, scientists discovered a single recessive mutation of *Arabidopsis thaliana* by mutant screening that abolished SA- or its analog-induced SAR-related gene expression and exhibited enhanced disease susceptibility, named *AtNPR1* [26], and also named *AtNIM1* [27] and *AtSAI1* [28]. In addition, AtNPR1 has roles beyond SA-induced defense responses, such as in rhizobacterium–triggered induced systemic resistance (ISR) [29], crosstalk between SA and jasmonic acid (JA) signaling pathways [30,31], and cold acclimation [32]. The *AtNPR1* gene encodes a protein with two conserved protein–protein interaction domains: broad complex, tramtrack, and bric-a-brac/pox virus and zinc finger (BTB/POZ) domain at the N-terminus and ankyrin repeats in the central region [33,34,35,36]. With the completion of the *Arabidopsis* genome sequence, there are five *AtNPR1* paralogs in *Arabidopsis* genome [37], named *AtNPR2*, *AtNPR3*, *AtNPR4*, *Arabidopsis BLADE-ON-PETIOLE2* (*AtBOP2*; also named *AtNPR5*), and *AtBOP1* (also named *AtNPR6*) [38,39,40]. Phylogenetic analysis divides the *AtNPR1*-like gene family into three functionally distinct clades. Each of the three clades contains two family members with functional redundancy. In the first clade, AtNPR1 and AtNPR2 are SA receptors and act as transcriptional co-activators in plant immunity [9,41]. In the second clade, AtNPR3 and AtNPR4 are also SA receptors and serve as transcriptional co-repressors in plant defense [9,42]. In the third clade, AtBOP1 and AtBOP2 are involved in plant growth and development [39,40].

In the absence of pathogen invasion, intracellular SA content is low and the NPR1 protein resides predominantly in the cytoplasm as an inactive oligomer formed via intermolecular disulfide bonds [43]. Moreover, NPR3 and NPR4, acting as transcriptional co-repressors, interact with TGA (TGACG motif-binding factor) transcription factors [42] to suppress transcription of SA-responsive genes in the nucleus [9]. In response to pathogen challenge, intracellular SA concentration rises rapidly, leading to conformational change of NPR1 from oligomer to monomer [43,44]. The monomeric NPR1 is then translocated into the nucleus mediated by its C-terminal bipartite nuclear localization signal (NLS) [45]. There, NPR1 binds SA and interacts with the same subset of TGAs to activate its transcriptional co-activator function [6,9]. Meanwhile, NPR1 recruits CDK8 (cyclin-dependent kinase 8) and WRKY (W-box-binding factor) transcription factors to the *NPR1* promoter to positively regulate its own expression [46,47]. Moreover, SA-binding to NPR3 and NPR4 eliminates their transcriptional co-repressor activity on TGAs [9]. This enables TGAs to turn on defense-related gene expression and activates defense response. In addition to pathogen invasion, exogenous application of SA or its analogs (BTH; benzothiodiazole and INA; 2,6-dichloroisonicotinic acid) could also induce the resistance mechanism in plants [48,49]. Taken together, NPR1, and its paralogues NPR3 and NPR4, are all SA receptors through an antagonistic manner to finely regulate plant immune response dependent on distinct threshold levels of SA [9].

*Arabidopsis NPR1* and its homologs have been proved to be involved in SA-mediated defense responses through genetic transformation in many plant species. For example, these *NPR1*-like genes likely function similar to *AtNPR1*, which acts as a positive regulator in the SA-mediated immunity. Transgenic rice (*Oryza sativa*) overexpressing *OsNPR1* displayed enhanced resistance to bacterial blight pathogen *Xanthomonas oryzae* pv. *oryzae* (*Xoo*) and fungal blast pathogen *Magnaporthe oryzae* (*Mo*), and knockdown of *OsNPR1* exhibited increased susceptibility to *Xoo* and *Mo* [50,51,52]. Transgenic mustard (*Brassica juncea*) overexpressing *BjNPR1* showed enhanced resistance to fungal pathogens *Alternaria brassicae* and *Erysiphe cruciferarum* [53]. Overexpression of mulberry (*Morus multicaulis*) *MuNPR1* in *Arabidopsis* displayed enhanced resistance to *Pseudomonas syringae* pv. *tomato* DC3000 (*Pst*DC3000) [54]. Knockdown of barley *HvNPR1* [55,56], tobacco *NtNPR1* [57], and tomato *NPR1*-like gene [58] showed elevated susceptibility to powdery mildew fungus, tobacco mosaic virus (TMV), and *Ralstonia solanacearum*, respectively. Moreover, these *NPR1*-like genes likely function similar to *AtNPR3*/*4*, which acts as a negative regulator in the SA-mediated resistance. Transgenic rice overexpressing *OsNPR2* and *OsNPR3* had no enhanced resistance to *Xoo* [51], and overexpression of *OsNPR3* using its own promoter resulted in increased expression of several *PR* genes and enhanced resistance only after treatment with BTH [59]. Knockdown of *Theobroma cacao TcNPR3* conferred resistance against *Phytophthora tropicalis* [60,61]. Overexpression of mulberry *MuNPR4* [54] or strawberry *FvNPRL-1* [62] in *Arabidopsis* both showed enhanced susceptibility to *Pst*DC3000.

In emmer wheat, *wNPR1* was isolated from *Triticum turgidum* ssp. *durum* by homology cloning strategy [63], and transgenic barley overexpressing *wNPR1* displayed elevated resistance to *Mo* [64]. Overexpression of *AtNPR1* or *Secale cereale ScNPR1* in bread wheat both conferred enhanced FHB resistance [65,66,67,68]. Based on the above research, it is necessary to systematically identify and analyze the *NPR1*-like family in bread wheat (*T. aestivum*) and its relatives (*T. urartu*, *T. dicoccoides*, and *Ae. tauschii*). The recently published reference genome sequences provide an opportunity for this study [69,70,71,72]. Here, the *NPR1*-like family was identified from bread wheat and its relatives’ genomes. These putative *NPR1*-like genes were analyzed in detail, including molecular characterization, chromosomal distributions, phylogenetic classification, gene structures, protein domain compositions, conserved motifs and amino acid residues, and cis-regulatory elements. The collinearity analysis for *NPR1*-like genes was performed among bread wheat and its relatives. Furthermore, the expression pattern of *TaNPR1*-like genes in various tissues/organs and under biotic stress conditions was also analyzed using publicly available bread wheat RNA-sequencing (RNA-seq) datasets. Overall, these results provide an invaluable resource for further functional study of *NPR1*-like genes in bread wheat and its relatives.

## 2. Results

### 2.1. Identification, Phylogeny, and Characterization of NPR1-Like Genes in Bread Wheat and its Relatives

Utilizing bioinformatics tools that contain BLAST search and HMMER analysis, a total of 40 putative *NPR1*-like genes were identified from bread wheat and its relatives’ genomes. Among them, there were 17 in *T. aestivum*, five in *T. urartu*, 12 in *T. dicoccoides*, and six in *Ae. tauschii* (Table 1). To study the phylogeny of the *NPR1*-like family, an unrooted phylogenetic tree was constructed using the sequences of the 40 NPRs, and 4 OsNPRs, 5 PaNPRs, and 31 NPRs obtained through molecular cloning techniques from eight monocots and 15 dicots in the published literature (Appendix A). Although individual phylogenetic analysis does not adequately perform functional annotation, phylogenetic grouping perhaps provides a reference for understanding functional diversification of the *NPR1*-like family. The results demonstrated that NPR1-like proteins were clustered into three major clades: clade I (AtNPR1/2 subfamily) containing OsNPR1 [50,51,52], BjNPR1 [53], MuNPR1 [54], HvNPR1 [55,56], and NtNPR1 [57], etc; clade II (AtNPR3/4 subfamily) containing OsNPR2 [51], OsNPR3 [59], TcNPR3 [60,61], and MuNPR4 [54], etc; and clade III (AtBOP1/2 subfamily) (Figure 1). In tetraploid and hexaploid wheat, homoeologous genes located in the branch ends of each clade belonging to the A, B, or D subgenomes, were regarded as the homeoalleles of one *NPR1*-like gene arising from allpolyploidization in genome evolution (Appendix A). All of the 40 identified *NPR1*-like genes were named and classified based on the phylogenetic relationship of the *Arabidopsis NPR1*-like family.

For example, 17 bread wheat *NPR1*-like genes were identified and evenly distributed in the three clades, consistent with the *NPR1*-like genes in *Arabidopsis* (Table 1). Each branch end of the *TaNPR1*-like family was composed of three homeoalleles in A, B, and D subgenomes, except the *TaNPR2* branch, which had two homeoalleles in A and D subgenomes. Clade I contained *TaNPR1-A*/*-B*/*-D* and *TaNPR2-A*/*-D*. *TaNPR3-A*/*-B*/*-D* and *TaNPR4-A*/*-B*/*-D* clustered in clade II. Clade III consisted of *TaNPR5-A*/-*B*/*-D* and *TaNPR6-A*/*-B*/*-D*. The *TaNPR1*-like genes were located on homoeologous group 3 chromosomes (*TaNPR1-A*/-*B*/*-D*, *TaNPR3-A*/-*B*/*-D*, and *TaNPR5-A*/-*B*/*-D*), homoeologous group 4 chromosomes (*TaNPR2-A* and *TaNPR4-A*/-*B*/*-D*), homoeologous group 6 chromosomes (*TaNPR5-A*/-*B*/*-D*), and homoeologous group 7 chromosomes (*TaNPR2-D*) (Appendix A). The sequence length of TaNPR proteins ranged from 487 (TaNPR2-D) to 618 (TaNPR3-A) amino acids. The average molecular weight was 59.87 kDa, varying between 51.62 kDa (TaNPR6-B) and 67.45 kDa (TaNPR3-A). The isoelectric points (pI) of TaNPR1-like members ranged from 5.18 (TaNPR2-A) to 6.12 (TaNPR5-A), with an average of 5.76, showing a weak acid. A single transcript was available in 10 of 17 *TaNPR1*-like genes, and the remaining seven genes had two splice variants.

### 2.2. Sequence and Structural Analysis of TaNPR1-Like Genes and Proteins

To further understand the potential functions of *TaNPR1*-like genes, the structural feature and sequence composition were analyzed using GSDS, NCBI-CDD, and the Clustal Omega program. The exon-intron distributions of *TaNPR1*-like genes were consistent with the corresponding phylogenetic clade in *Arabidopsis* (Figure 2A). Clade I, *TaNPR1-A*/*-B*/*-D* and *TaNPR2-A*/*-D*, and clade II, *TaNPR3-A*/*-B*/*-D* and *TaNPR4-A*/*-B*/*-D*, contained four exons and three introns. Clade III, *TaNPR5-A*/*-B*/*-D* and *TaNPR6-A*/*-B*/*-D*, had two exons and one intron. The protein domain composition revealed that all 17 TaNPR proteins contained an N-terminal BTB/POZ domain and ANK repeats in the central region similarly to AtNPR1 (Figure 2B). However, only clade I, TaNPR1-A/-B/-D and TaNPR2-A/-D, and clade II, TaNPR3-A/-B/-D and TaNPR4-A/-B/-D, contained the NPR1-like C-terminal region that was essential for AtNPR1 activity [35,36]. The C-terminal region contained penta-amino acid motif (LENRV), NIMIN-binding region, and nuclear localization signal (NLS) (Figure 2C) [45,73].

Multiple sequence alignments were performed to examine the conservation of residues, motifs, and domains in TaNPRs and known-function NPRs (AtNPR1 to AtNPR6, OsNPR1, and wNPR1) (Figure 2C). The results revealed that npr1-1 (His334Tyr), npr1-2 (Cys150Tyr) [33], and nim1-2 (His300Tyr) [74] point mutations in AtNPR1 were completely conserved in all 17 TaNPR1-like proteins. Moreover, nim1-4 (Arg432Lys) [74] in AtNPR1 and npr4-4D (Arg419Gln) [9] in AtNPR4 mutant sites were also conserved in clade I, TaNPR1-A/-B/-D and TaNPR2-A/-D, and clade II, TaNPR3-A/-B/-D and TaNPR4-A/-B/-D. Three cysteine residues (C82, C216, and C156) in AtNPR1 [43,44] that participated in its oligomer–monomer transition were also highly conserved in all 17 TaNPR1-like proteins, except C82 in TaNPR2-A/-D. The Arg432 residues in AtNPR1, Arg428 in AtNPR3, and Arg419 in AtNPR4 required for their perception of SA [9] were conservative among clade I, TaNPR1-A/-B/-D and TaNPR2-A/-D, and clade II, TaNPR3-A/-B/-D and TaNPR4-A/-B/-D. In addition, EAR-like motif (VDLNETP) required for AtNPR3/4 as a co-repressor factor [9] was also present in clade II, TaNPR3-A/-B/-D and TaNPR4-A/-B/-D. Therefore, *TaNPR1*-like genes belonging to the same *Arabidopsis* subfamily have similar gene structures and functional domain composition, as well as conserved motifs and amino acid residues, deducing that their orthologs probably display similar biological functions in bread wheat.

### 2.3. Analysis of cis-Regulatory Elements in the Promoter Regions of TaNPR1-like Genes

In plants, cis-regulatory elements control the expression of target genes by interacting with transcription factors [75]. Therefore, identifying potential cis-regulatory elements in the promoter regions of *TaNPR1*-like genes will provide useful information for understanding their regulatory expression. The 1000-bp upstream promoter sequences of *TaNPR1*-like genes were selected and submitted to the PlantCARE online service [76] for SA-responsive cis-regulatory identification.

The results showed that two hormone-related regulatory elements, CAAT-box and TATA-box, were overrepresented in all 17 *TaNPR1*-like gene promoters (Figure 3). The *activation sequence-1* (*as-1*) element (TGACG) involved in transcription activation of several SA-regulated *PR* genes was detected in all *TaNPR1*-like gene promoters, except *TaNPR1-B* and *TaNPR5-A*/*-B*/*-D*. Moreover, the W-box element (TTGACC) was known to be the DNA-binding site for the SA-induced WRKY transcription factors [46]. This W-box motif enriched in the *AtNPR1* and *OsNPR1* promoter was only found in the *TaNPR1-A*/*-B*/*-D* promoter, suggesting that *TaNPR1-A*/*-B*/*-D* may be directly regulated by WRKYs in a manner similar to *AtNPR1* [47].

### 2.4. Chromosomal Distribution and Collinearity Analysis of NPR1-like Genes among Bread Wheat and Its Relatives

To explore the orthologous relationships of *NPR1*-like genes among bread wheat and its relatives, 6 *Ae. tauschii*-NPRs and 6 *T. aestivum*-NPRs in the D subgenome, 12 *T.dicoccoides*-NPRs and 11 *T. aestivum*-NPRs in the AB subgenome, and 5 *T.urartu*-NPRs and 7 *T.dicoccoides*-NPRs in the A subgenome protein sequences were used to generate phylogenetic trees (Appendix A). Pairs of six *Aet*/*Ta*-D, 11 *Td*/*Ta*-AB, and four *Tu*/*Td*-A orthologs were identified (Appendix A) and mapped to genome chromosomes (Figure 4).

The collinearity analysis illustrated that six pairs of *Aet*/*Ta*-D orthologs were located on the same chromosomes with three on 3D, one on 4D, one on 5D, and one on 7D (Figure 4). Moreover, 10 *Td-NPRs* could be mapped to bread wheat AB subgenomes on the same chromosomes with three on 3A, one on 4A, one on 5A, three on 3B, one on 4B, and one on 5B, except *TdNPR2-A-2*/*TaNPR2-A* orthologous gene pairs (*TdNPR2-A-2* on 7AS, *TaNPR2-A* on 4AL). Furthermore, four *Tu*/*Td*-A orthologous gene pairs were located on the same chromosomes, with three on 3A and one on 4A. However, orthologous *TuNPR4* was located on *T.urartu* chromosome 4AS, corresponding to *TdNPR4-A* on *T.dicoccoides* chromosome 4AL. The orthologous gene pairs on different chromosomal positions indicated that chromosomal recombination events (chiasmata and crossing-over) had occurred during the evolutionary process. On the whole, however, the *NPR1*-like genes had an intact collinearity among bread wheat and its relatives, suggesting that the evolution of this gene family was more conservative following formation of hexaploid wheat.

### 2.5. Expression Analysis of TaNPR1-Like Genes in Various Tissues/Organs

To analyze the tissue/organ-specific expression patterns of *TaNPR1*-like genes, the expression data in eight tissues/organs (roots, stem axis, first leaf blade, shoot apical meristem, flag leaf, internode, glumes, and lemma) during the seedling and reproductive stages [77] were downloaded from the wheat URGI website.

The results revealed that *TaNPR1*-like genes were constitutively expressed in various tissues/organs, although the level of expression varied greatly (Figure 5). The expression level of *TaNPR1-A*/*-B*/*-D* was moderate and was relatively high in root and flag leaf tissues. The expression of *TaNPR2-A*/*-D* was much lower in most tissues/organs with a similar pattern as *AtNPR2* in *Arabidopsis* tissues [78]. *TaNPR3-A*/*-B*/*-D* and *TaNPR4-A*/*-B*/*-D* had significantly higher expression in this family, showing relatively balanced expression in eight tissues. *TaNPR5-A*/*-B*/*-D* and *TaNPR6-A*/*-B*/*-D* had strong expression levels in the shoot apical meristem at the seeding stage and in internode, glume, and lemma tissues at the reproductive stage, but were weakly expressed in other tissues/organs (roots, stem axis, first leaf blade, and flag leaf). The results infer that *TaNPR5-A*/*-B*/*-D* and *TaNPR6-A*/*-B*/*-D* appear to be similarly related in growth and development to *AtBOP1*/*2* [39,40].

### 2.6. Expression Analysis of TaNPR1-Like Genes in Response to Biotic Stress

To determine the potential functions of *TaNPR1*-like genes in response to biotic stress, the expression data under PAMP (chitin and flg22) treatment [77] and inoculation with *F. graminearum* [79] were obtained from the wheat URGI website.

In leaf tissue treated with fresh water/mock or PAMPs (chitin and flg22), the expression levels of *TaNPR2-A*/*-D*, *TaNPR5-A*/*-B*/*-D*, and *TaNPR6-A*/*-B*/*-D* were much lower (TPM < 0.3) than those of other family members (Figure 6A). The transcript levels of *TaNPR1-A*/*-B*/*-D*, *TaNPR3-A*/*-B*/*-D*, and *TaNPR4-A*/*-B*/*-D* were up-regulated after PAMP (chitin and flg22) treatment for 30 min compared with the mock group. In particular, the up-regulation extent of *TaNPR1-A*/*-B*/*-D* significant increased at 30 min. Additionally, the expression of *TaNPR1-A*/*-B*/*-D*, *TaNPR3-A*/*-B*/*-D*, and *TaNPR4-A*/*-B*/*-D* was not significantly different between control and experimental groups at 180 min. The results indicate that *TaNPR1-A*/*-B*/*-D*, *TaNPR3-A/-B/-D*, and *TaNPR4-A/-B/-D* could possibly partake in bread wheat early basal resistance.

In spike tissue inoculated with mock or *F. graminearum*, *TaNPR2-A*/*-D* transcripts showed the lowest level (TPM < 0.2) in this family (Figure 6B). Only *TaNPR1-A*/*-B*/*-D* was up-regulated at 30 and 50 h after inoculation (hai) compared with the mock group. There was no significant difference in *TaNPR3-A*/*-B*/*-D* and *TaNPR4-A*/*-B*/*-D* between control and experimental groups at 30 and 50 hai. Moreover, *TaNPR5-A*/*-B*/*-D* and *TaNPR6-A*/*-B*/*-D* were slightly down-regulated at 30 and 50 hai. The results imply that *TaNPR1-A*/*-B*/*-D* may have similar functions as *AtNPR1* and *ScNPR1* in response to *F. graminearum* in bread wheat [65,66,67,68].

## 3. Discussion

### 3.1. Identification and Phylogenic Analysis of NPR1-Like Family

In this study, we isolated five *NPRs* in diploid *T. urartu*, six in diploid *Ae. tauschii*, 12 in tetraploid emmer wheat (*T. dicoccoides*), and 17 in hexaploid wheat (*T. aestivum*) by genome-wide approaches. The *NPR1*-like family is a small community in triticeae crops similar to other plant species, such as four members in *Oryza sativa* [51], six in *Arabidopsis* [40], five in *Persea Americana* [79], six in *Populus trichocarpa* [80], three in *Vitis vinifera* [80], and four in *Medicago truncatula* [80]. The phylogenetic tree showed that the 40 identified *NPR1*-like genes in bread wheat and its relatives clustered into three major clades and distributed evenly among the clades (Figure 1). For example, *NPR1* homologs in *Ae.tauschii* and *T.aestivum* divide into three clades and each clade contains two family members. Furthermore, the phylogenetic analysis of *NPR1*-like family further supports the previously evolutionary hypothesis [78]. It reveals that an ancestral gene of *NPR*1-like family through duplication differentiates into two main clades, *NPR* and *BOP*. Then, the progenitor *NPR* gene could undergo a second round of duplication event leading to the *NPR1*/*2* and the *NPR3*/*4* clades. Since the above-mentioned five monocots and five dicots have at least one member in each of the three clades (Figure 1) [78], the ancient duplication events resulting in functional divergence of *NPR1*-like genes likely occurred prior to the monocot-dicot split. After the monocot-dicot split, members of the three clades may have experienced another round of gene duplication events, leading to the current state of each clade with two related genes, such as six *NPRs* (*AtNPR1*/*AtNPR2*, *AtNPR3*/*AtNPR4*, and *AtBOP1*/*AtBOP2*) in *Arabidopsis* and six *NPRs* (*AetNPR1*/*AetNPR2*, *AetNPR3*/*AetNPR4*, and *AetNPR5*/*AetNPR6*) in *Ae. tauschii* (Figure 1).

### 3.2. Sequence and Structural Features of TaNPR1-Like Genes and Proteins

The sequence and structural features of *TaNPR1*-like genes further support the phylogenetic analysis. *NPR1* homologs on the same clade of bread wheat and *Arabidopsis* share similar exon-intron structures (Figure 2A). Moreover, the structural organization of *NPR1*-like genes in clade I and II consisted of four exons and three introns, which were also conserved in other plant species, such as rice *OsNPR1*, *OsNPR2*, and *OsNPR3* [51], barley *HvNPR1* [55], *Theobroma cacao TcNPR1* and *TcNPR3* [61], and *Persea americana PaNPR1, PaNPR2*, and *PaNPR4* [80]. This conservation across different species showed that the *NPR1*-like family was also conserved in the genomic structure.

On the other hand, all of the 17 TaNPR1-like proteins harbor the BTB/POZ and ANK repeat domains (Figure 2B). Only TaNPR1-A/-B/-D and TaNPR2-A/-D in clade I, and TaNPR3-A/-B/-D and TaNPR4-A/-B/-D in clade II, contain an NPR1-like C-terminal region. In particular, the Arg432 residues in AtNPR1, Arg428 in AtNPR3, and Arg419 in AtNPR4 required for binding SA were highly conserved in TaNPR1-A/-B/-D, TaNPR2-A/-D, TaNPR3-A/-B/-D, and TaNPR4-A/-B/-D (Figure 2C). Moreover, the transcriptional repression motif (VDLNETP) in the AtNPR3/4 C-terminal domain was also present in TaNPR3-A/-B/-D and TaNPR4-A/-B/-D. The results demonstrate that TaNPR1-A/-B/-D, TaNPR2-A/-D, TaNPR3-A/-B/-D, and TaNPR4-A/-B/-D may be SA receptors. TaNPR1-A/-B/-D and TaNPR2-A/-D likely act as transcriptional co-activators, and TaNPR3-A/-B/-D and TaNPR4-A/-B/-D likely act as transcriptional co-repressors to participate in SA-induced immunity in bread wheat.

### 3.3. Evolution and Expansion of NPR1-Like Family among Bread Wheat and Its Relatives

The gene duplication events deriving from tandem duplication, segmental duplication, and genome-wide polyploidization cause gene family expansion in plant genome evolution [81]. Chromosome distribution and collinearity analysis indicated that allopolyploid events were the main reason for the expansion of *NPR1*-like family in hexaploid wheat (Figure 4). The first polyploidy event of hybridization between *T. urartu* and *A. speltoides* resulted in the *NPR1*-like family being duplicated in *T. turgidum*. Subsequently, the second polyploidy event, which crossed tetraploid emmer wheat and *Ae. tauschii*, led to the 17 *NPR1*-like sequences in bread wheat, including six, five, and six genes from the A, B, and D subgenomes, respectively.

Although the *NPR1*-like family was conservative among bread wheat and its relatives, intrachromosomal serial replication and gene-loss events also occurred during the evolution process. For example, replication events appeared in emmer wheat *TdNPR2-A* and *T. urartu TuNPR5*, resulting in *TdNPR2-A-1*/*TdNPR2-A-2* and *TuNPR5-1*/*TuNPR5-2*. Gene-loss events occurred in *NPR2* on chromosome 7B of bread wheat and emmer wheat, and on chromosome 7A of *T. urartu*, and in *NPR6* on chromosome 5A of *T. urartu*. To find the residual sequences of the missing *NPR1*-like genes, putative homologous chromosomal regions were identified between emmer wheat and *T. urartu* using MCScanX [82]. In the collinear chromosomal region containing *TdNPR2-A*, we manually found a gene (TuG1812G0700000282.01) on chromosome 7AL of *T. urartu* (Figure 4). The protein length (396 amino acids) translated by the gene is smaller than that of *TdNPR2-A* (570 amino acids). Moreover, this protein also contains ANK repeats and an NPR1-like C-terminal domain similar to TdNPR2-A, but its N-terminal BTB/POZ domain is unintegrated. Furthermore, *TaNPR2-A/D* have very low levels of expression both in normal tissues/organs and in stress-treated samples (Figure 5 and Figure 6), and the rice *NPR1*-like family has only one member (*OsNPR1*) in clade I (Figure 1). It is supposed that these functionally redundant genes may have been subjected to gradual pseudogenization or gene deletion during evolution.

### 3.4. Functional Divergence of TaNPR1-Like Genes

The tissue/organ-specific expression patterns usually reflect their corresponding biological functions. From in silico assessment of RNA-seq experiments, *TaNPR5-A*/*-B*/*-D* and *TaNPR6-A*/*-B*/*-D* in clade III exhibited specific expression in the shoot apical meristem, internode, glume, and lemma tissues (Figure 5), suggesting they may participate in growth and development.

The expression profiles of *TaNPR1*-like genes upon biotic stresses were also investigated to examine *TaNPR1*-like gene functions (Figure 6). Functional receptors on the plant cell membranes recognize PAMPs, which trigger the first line of defense, called PTI. SA binds to its receptors (NPR1, NPR3, and NPR4) to induce the expression of *PR* genes, which contribute to trigger immune responses [5]. Under mock or PAMP (chitin and flg22) treatments, the up-regulation of *TaNPR1-A*/*-B*/*-D*, *TaNPR3-A*/*-B*/*-D*, and *TaNPR4-A*/*-B*/*-D* in 17 *NPR1*-like members was significant at 30 min between mock and PAMP treatment groups, and there was no significant difference at 180 min (Figure 6A). Moreover, past studies have shown that overexpression of *AtNPR1* or *ScNPR1* in the transgenic bread wheat line enhances defense against *F. graminearum* [65,66,67,68]. Under mock or *F. graminearum* treatments, only *TaNPR1-A*/*-B*/*-D* was up-regulated at 30 and 50 hai (Figure 6B). Taken together, these putative candidates could be used as the preferred genes to prove their biological functions through molecular experiments in development and defense. 

## 4. Methods

### 4.1. Sources of Sequence Data

The genome sequences of *T. aestivum* (*Ta*), *T. dicoccoides* (*Td*), and *Ae. tauschii* (*Aet*) were obtained from Ensembl Plants (http://plants.ensembl.org/index.html), and *T. urartu* (*Tu*) sequence was acquired from MBKbase (http://www.mbkbase.org). The NPR1-related protein sequences of *Oryza sativa* (*Os*), *Hordeum vulgare* (*Hv*), *Brachypodium distachyon* (*Bd*), *Zea mays* (*Zm*), *T. dicoccoides* (*w*), *Musa acuminate* (*M*), *Gladiolus hybridus* (*Gh*), *Lilium* (*LhSor*), *Arabidopsis thaliana* (*At*), *Brassica juncea* (*Bj*), *Morus multicaulis* (*Mu*), *Nicotiana tabacum* (*Nt*), *Gossypium hirsutum* (*Gh*), *Glycine max* (*Gm*), *Malus domestica* (*Mp*), *Vitis vinifera* (*Vv*), *Solanum lycopersicum* (*Le*), *Theobroma cacao* (*Tc*), *Carica papaya* (*Cp*), *Ipomoea batatas* (*Ib*), *Populus deltoids* (*Pd*), *Pyrus pyrifolia* (*Pp*), and *Persea Americana* (*Pa*) were downloaded from the GenBank database (https://www.ncbi.nlm.nih.gov/genbank/). All sequence information is listed in Appendix A.

### 4.2. Identification of NPR1-Like Genes

To identify *NPR1*-like genes in bread wheat and its relatives (*T. urartu*, *T. dicoccoides*, and *Ae. tauschii*), three bioinformatics methods were executed, namely, BLASTP search, HMMER analysis, and validation of conservative domains. Firstly, a local protein database of bread wheat was established using the basic local alignment search tool (BLAST, ftp://ftp.ncbi.nlm.nih.gov/blast/executables/blast+/LATEST/). All the *Arabidopsis* and rice NPR1-like protein sequences were used as queries to perform BLASTP searches against the local protein database (E-value < 1 × 10^−5^). Secondly, the hidden markov model (HMM) profile of six AtNPRs and four OsNPRs was constructed to search the protein database using HMMER3.0 (http://hmmer.org/). After removing the redundant sequences of the above two search results, a total of 120 candidate sequences were retrieved and verified for conserved domains using the NCBI conserved domain database (CDD, https://www.ncbi.nlm.nih.gov/cdd), Pfam database(https://pfam.xfam.org/), and InterPro database (http://www.ebi.ac.uk/interpro/). The candidate proteins without N-terminal BTB/POZ domain and ANK repeats in the central region were removed. Finally, a total of 20 putative NPR1-like protein sequences translated from 17 genes were identified from bread wheat genome. Using the same method, the *NPR1*-like genes of *T.urartu*, *Ae. tauschii*, and *T.dicoccoides* were retrieved from their genomic databases.

### 4.3. Analysis of NPR1-Like Gene Characteristics

Protein properties of identified *NPR1*-like genes, including protein length, molecular weights, and isoelectric points, were estimated using ExPASy (https://web.expasy.org/protparam/). Exon-intron structures of *NPR1*-like genes in bread wheat and *Arabidopsis* were displayed using the Gene Structure Display Server (GSDS, http://gsds.cbi.pku.edu.cn/) [83]. NCBI-CDD was used to identify the conservative domain compositions of 17 TaNPRs and 6 AtNPRs proteins, and the results were visualized by the TBtools software [84]. Multiple sequence alignments of TaNPRs and known-function NPRs were generated using the Clustal Omega program (https://www.ebi.ac.uk/Tools/msa/clustalo/) and visualized in Jalview Version 2 [85]. The promoter sequences (1000-bp upstream of the ATG translation start codon) of *TaNPR1*-like genes were extracted from the bread wheat genome sequence. cis-Regulatory elements were predicted in the PlantCARE database (http://bioinformatics.psb.ugent.be/webtools/plantcare/html/) [76]. The promoter sequences are listed in Appendix A.

### 4.4. Phylogenetic Tree Construction, Chromosomal Location, and Homologous Relationships

To investigate the phylogenetic relationship of *NPR1*-like genes, the full-length protein sequences were aligned using ClustalW. Subsequently, an unrooted phylogenetic tree was constructed by MEGA v7.0 using the neighbor-joining (NJ) algorithm with the following parameters: bootstrap method (1000 replicates), Poisson model, and complete deletion. Chromosomal locations of *NPR1*-like genes in bread wheat and its relatives were obtained from general feature format (GFF3) files. MapChart software was used to map the distribution of *TaNPR1*-like genes. Phylogenetic trees for the *T. aestivum* AB subgenome-*T. dicoccoides*, *T. aestivum* D subgenome-*Ae. tauschii*, and *T. dicoccoides* A subgenome-*T. urartu* NPR1-like proteins were constructed in MEGA v7.0. In the phylogenetic trees, two genes from distinct species located in the same branch were defined as orthologs [86]. The orthologous gene pairs among bread wheat and its relatives were identified based on the homologous relationships (Appendix A). Subsequently, the Circos tool [87] was used to display the chromosomal locations and homologous relationships of *NPR1*-like family among bread wheat and its relatives.

### 4.5. Expression Profiles of TaNPR1-Like Genes in RNA-Seq

To study the expression profiling of *TaNPR1*-like genes in different tissues/organs and under stress conditions, three sets of publicly available transcriptome data were downloaded from the wheat URGI (https://wheat-urgi.versailles.inra.fr/Seq-Repository/Expression). Raw data of the three datasets was also deposited in the NCBI Short Read Archive (SRA) database under accession numbers PRJEB25639 [77], ERS1978239 [77], and ERP003465 [79].

The PRJEB25639 data were collected from various tissues/organs of spring wheat cultivar Azhurnaya, such as roots, stem axis, first leaf blade, and shoot apical meristem at the seeding stage; and flag leaf blade, internode, glumes, and lemma at the reproductive stage. The ERS1978239 data were generated from leaf samples of three-week-old Chinese Spring (CS) after fresh water/mock or PAMP (chitin and flg22) treatments for 30 min and 180 min. The ERP003465 data were obtained from mature inflorescences of the FHB-resistant spring wheat line CM-82036 after mock or *F. graminearum* treatments for 30 and 50 h. For all the aforementioned transcriptome datasets, three biological replicates were in each treated sample. The expression levels of *TaNPR1*-like genes were quantified as transcripts per kilobase million (TPM).

## 5. Conclusions

In summary, a total of 40 *NPR1*-like genes were identified from bread wheat and its relatives’ (*T. urartu*, *T. dicoccoides*, and *Ae. tauschii*) genomes. The *TaNPR1*-like genes were analyzed in depth, comprising molecular characterization, chromosomal distributions, phylogenetic classification, gene structures, protein domain compositions, conserved motifs, and amino acid residues, as well as cis-regulatory elements. A good collinearity of *NPR1*-like genes was present among bread wheat and its relatives. Based on RNA-seq data, *TaNPR1*-like genes exhibited distinct tissue/organ specific expression patterns and *TaNPR1-A*/*-B*/*-D*, *TaNPR3-A*/*-B*/*-D*, and *TaNPR4-A*/*-B*/*-D* were induced under biotic stress conditions. These results will be helpful in designing experiments to determine the biological functions and understanding the evolutionary relationship of the *NPR1*-like gene family in bread wheat and its relatives.

## Figures and Tables

**Figure 1 ijms-20-05974-f001:**
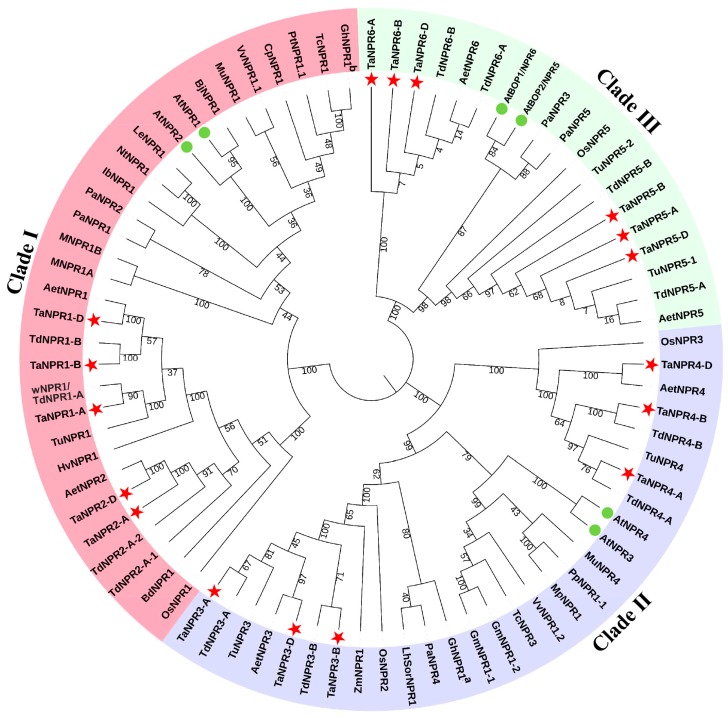
Phylogenetic analysis of NPR1 homolog proteins from different plant species. The tree was generated with MEGA v7.0 using the neighbor-joining (NJ) algorithm. NPR1-related proteins from eight monocot species *Oryza sativa* (*Os*), *Hordeum vulgare* (*Hv*), *Brachypodium distachyon* (*Bd*), *Zea mays* (*Zm*), *T. dicoccoides* (*w*), *Musa acuminate* (*M*), *Gladiolus hybridus* (*Gh*), *Lilium* (*LhSor*), and 15 dicot species *Arabidopsis thaliana* (*At*), *Brassica juncea* (*Bj*), *Morus multicaulis* (*Mu*), *Nicotiana tabacum* (*Nt*), *Gossypium hirsutum* (*Gh*), *Glycine max* (*Gm*), *Malus domestica* (*Mp*), *Vitis vinifera* (*Vv*), *Solanum lycopersicum* (*Le*), *Theobroma cacao* (*Tc*), *Carica papaya* (*Cp*), *Ipomoea batatas* (*Ib*), *Populus deltoids* (*Pd*), *Pyrus pyrifolia* (*Pp*), *Persea Americana* (*Pa*). All the GenBank accession numbers are listed in Appendix A. ^a^
*Gladiolus hybridus*, ^b^
*Gossypium hirsutum*, *T. aestivum* (*Ta*), *T. urartu* (*Tu*), *T. dicoccoides* (*Td*), *Ae. tauschii* (*Aet*). Three major clades are distinguished with three colors, and NPRs from *T. aestivum* and *Arabidopsis* are labeled with red and green markers.

**Figure 2 ijms-20-05974-f002:**
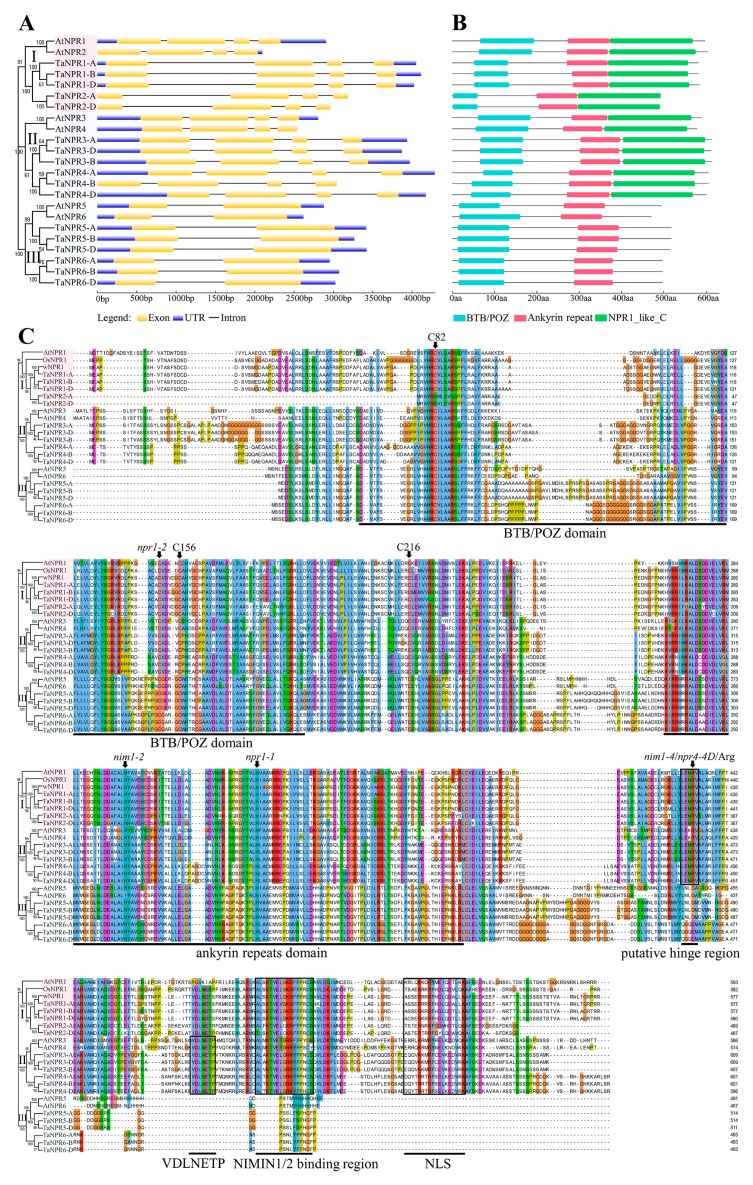
Gene structure and protein sequence comparison of *TaNPR1*-like genes with other known *NPR1*-like sequences. Exon-intron structures (**A**) of *NPR1*-like genes in *Arabidopsis* and *T. aestivum*. Exons, UTRs, and introns are denoted by yellow boxes, blue boxes, and grey lines, respectively. Conserved domain organization and distribution (**B**) of the BTB/POZ, ANK, and NPR1-like C-terminal region in *Arabidopsis* and *T. aestivum* NPR1-like proteins. Multiple alignment of amino acid sequences (**C**) of *T. aestivum* NPR1-like proteins (TaNPR1 to TaNPR6) and other NPR1-related proteins with experimentally confirmed functions (AtNPR1 to AtNPR6, OsNPR1, and wNPR1). The position of point mutation sites (npr1-1, npr1-2, nim1-2, and nim1-4 in AtNPR1), and three of the conserved cysteines residues (C82, C216, and C156 in AtNPR1) are marked with arrows. The conserved domains, BTB/POZ and ANK, and important motifs, putative hinge region (LENRV), EAR-like repression motif (VDLNETP), NIMIN-binding region, and nuclear localization signal (NLS), are highlighted with solid lines.

**Figure 3 ijms-20-05974-f003:**
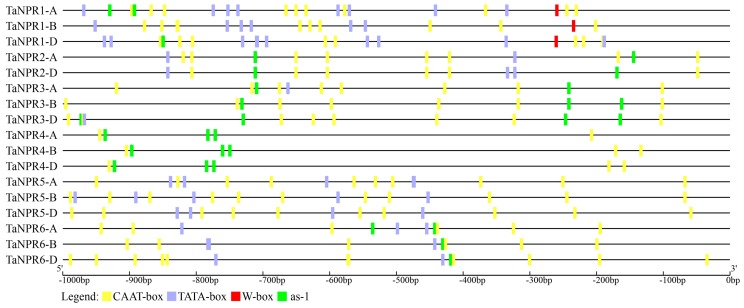
Analysis of potential cis-acting elements in the promoter regions of *TaNPR1*-like genes. The promoter regions (1000-bp upstream of the ATG translation start codon) of *TaNPR1*-like genes were used to analyze four specific SA-responsive cis-acting regulatory elements, including CAAT-box (yellow blocks), TATA-box (blue blocks), W-box (red blocks), and *as*-1 element (green blocks).

**Figure 4 ijms-20-05974-f004:**
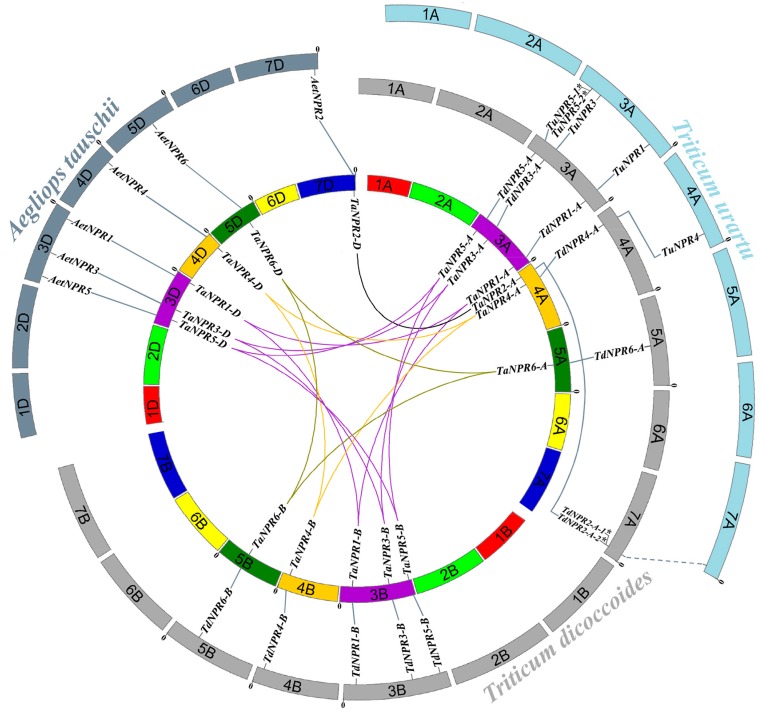
Chromosomal distribution and collinearity analysis of *NPR1*-like genes among bread wheat and relatives. Seven homologous groups of bread wheat chromosomes are represented by different colors. The genomes of *Ae. tauschii*, *T. dicoccoides*, and *T. urartu* surround the central *T. aestivum*. The *NPR1*-like genes are labeled according to their positions on the chromosomes. Homeoalleles of each *TaNPR1*-like gene are linked by lines with the corresponding color, except *TaNPR2*, which is connected by black lines since it is not located on the same subgenome. The collinearity of orthologous gene pairs among bread wheat and its relatives are displayed in the gray lines. A dotted line represents a possibly missing orthologous gene pair. Tandem duplication is indicated by asterisks.

**Figure 5 ijms-20-05974-f005:**
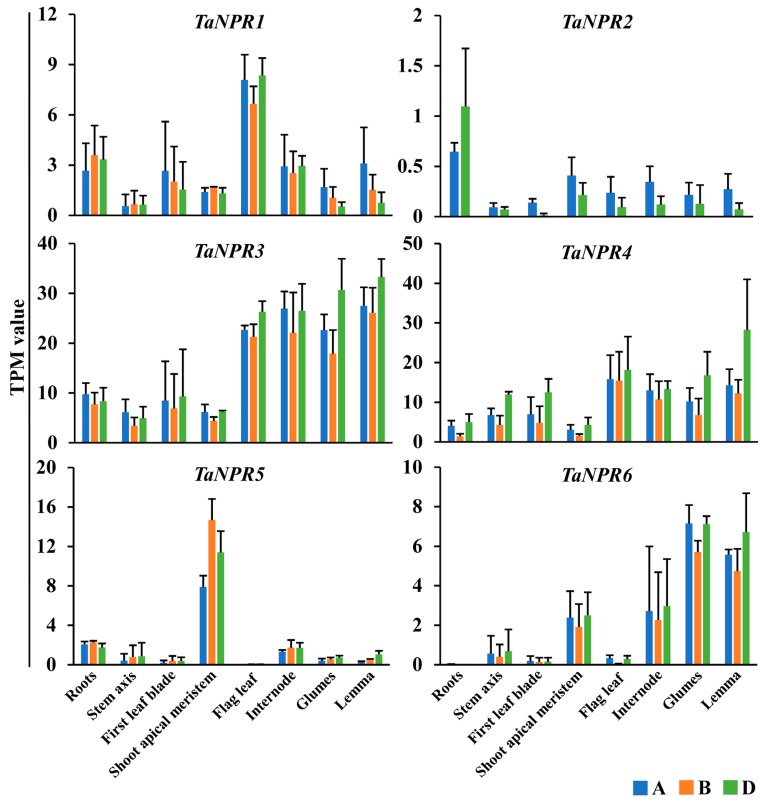
Expression profiles of *TaNPR1*-like genes across various tissues/organs by RNA-seq. The bread wheat tissues/organs at the seeding stage (roots, stem axis, first leaf blade, and shoot apical meristem) and reproductive stage (flag leaf blade, internode, glumes, and lemma) are marked on the horizontal axis. The unit of the y-axis is transcripts per kilobase million (TPM), and each TPM value of *TaNPR1*-like genes is the average of three biological replicates; error bars represent the SD.

**Figure 6 ijms-20-05974-f006:**
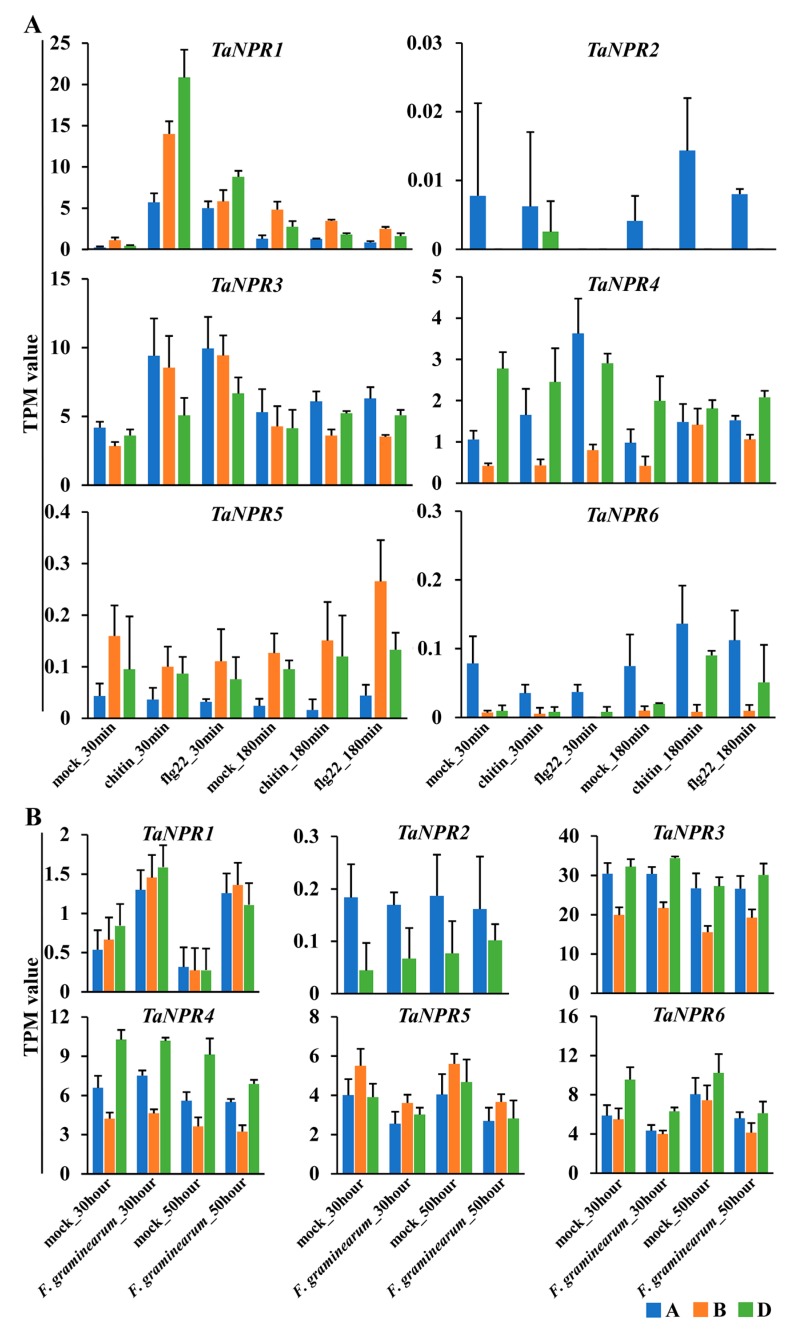
Expression pattern for *TaNPR1*-like genes under stress conditions by RNA-seq. Three-week-old leaf sections of Chinese Spring wheat plants treated with fresh water/mock or PAMPs (chitin and flg22) after 30 and 180 min (**A**). Mature inflorescences of the FHB-resistant CM-82036 treated with mock or *Fusarium graminearum* at 30 and 50 h after inoculation (hai) (**B**). The unit of the y-axis is transcripts per kilobase million (TPM), and each TPM value of *TaNPR1*-like genes is the average of three biological replicates; error bars represent the SD.

**Table 1 ijms-20-05974-t001:** Information about the *NPR1*-like genes in bread wheat and its relatives. Notes: AA, amino acid sequence length; MW, molecular weight; pI, isoelectric point; Splicing, alternative splicing number; a bar (-) represents only a single transcript (no splice variants); *TdNPR1-A* could also be named *wNPR1* (98.9% identity), discovered by Cantu et al. [63].

Species	Clade	Gene Name	Sequence ID	Chromosome Location	AA	Mw (kDa)	pI	Splicing
*T.aestivum*	I	*TaNPR1-A*	TraesCS3A02G105400.1	3A:69290641-69294697(+)	577	63.57	5.37	-
		*TaNPR1-B*	TraesCS3B02G123800.1	3B:96812311-96816428(+)	577	63.83	5.49	2
		*TaNPR1-D*	TraesCS3D02G107500.1	3D:61072307-61076336(+)	580	63.78	5.27	-
		*TaNPR2-A*	TraesCS4A02G470500.1	4A:731397493-731400686(–)	489	54.91	5.18	-
		*TaNPR2-D*	TraesCS7D02G019000.1	7D:8353564-8356531(+)	487	54.49	5.88	-
	II	*TaNPR3-A*	TraesCS3A02G298800.2	3A:532934212-532938152(–)	618	67.45	5.6	2
		*TaNPR3-B*	TraesCS3B02G337700.1	3B:544108681-544112656(+)	609	66.45	5.59	2
		*TaNPR3-D*	TraesCS3D02G302900.1	3D:417737897-417741772(+)	607	66.31	5.45	-
		*TaNPR4-A*	TraesCS4A02G294400.1	4A:595817163-595821455(+)	601	66.61	5.82	-
		*TaNPR4-B*	TraesCS4B02G018900.2	4B:13944552-13947600(+)	601	66.90	5.8	2
		*TaNPR4-D*	TraesCS4D02G017500.1	4D:7769189-7773368(–)	596	66.39	5.86	-
	III	*TaNPR5-A*	TraesCS3A02G489000.1	3A:716756231-716759651(+)	514	54.19	6.12	-
		*TaNPR5-B*	TraesCS3B02G537400.1	3B:777578202-777581471(+)	514	54.20	6.06	-
		*TaNPR5-D*	TraesCS3D02G484100.1	3D:581113138-581116563(–)	511	53.78	6.03	-
		*TaNPR6-A*	TraesCS5A02G134700.2	5A:304537866-304540822(–)	493	51.65	6.11	2
		*TaNPR6-B*	TraesCS5B02G133700.1	5B:250233025-250236102(–)	493	51.62	6.11	2
		*TaNPR6-D*	TraesCS5D02G139600.1	5D:222737713-222740742(+)	493	51.72	6.11	2
*T.urartu*	I	*TuNPR1*	TuG1812G0300001179.01.T01	3A:64494723-64498668(+)	577	63.49	5.37	-
	II	*TuNPR3*	TuG1812G0300003503.01.T01	3A:544372634-544376515(+)	609	66.35	5.52	2
		*TuNPR4*	TuG1812G0400000243.01.T01	4A:12320155-12324208(–)	600	66.54	5.82	2
	III	*TuNPR5-1*	TuG1812G0300005248.01.T01	3A:705429640-705432414(+)	514	54.19	6.06	-
		*TuNPR5-2*	TuG1812G0300005239.01.T01	3A:704992825-704995600(+)	531	58.25	9.48	2
*T.dicoccoides*	I	*TdNPR1-A*	TRIDC3AG012960.1	3A:64964238-64967906(+)	580	63.81	5.37	-
		*TdNPR1-B*	TRIDC3BG017420.1	3B:105475063-105478776(+)	580	63.94	5.43	8
		*TdNPR2-A-1*	TRIDC7AG001960.1	7A:5808627-5811849(+)	570	63.86	5.88	5
		*TdNPR2-A-2*	TRIDC7AG002040.2	7A:6071183-6076497(–)	549	61.66	5.57	5
	II	*TdNPR3-A*	TRIDC3AG044830.2	3A:548025257-548029459(+)	609	66.37	5.45	5
		*TdNPR3-B*	TRIDC3BG050750.2	3B:556623106-556626770(+)	609	66.45	5.59	5
		*TdNPR4-A*	TRIDC4AG045270.3	4A:588643107-588646338(+)	601	66.61	5.82	3
		*TdNPR4-B*	TRIDC4BG003310.1	4B:13063896-13067432(+)	531	59.12	5.45	5
	III	*TdNPR5-A*	TRIDC3AG068800.5	3A:714043376-714046300(+)	516	54.33	6.13	9
		*TdNPR5-B*	TRIDC3BG078000.6	3B:789380796-789383751(+)	516	54.33	6.04	9
*T.dicoccoides*	III	*TdNPR6-A*	TRIDC5AG022110.4	5A:290229998-290232860(+)	493	51.68	6.11	5
		*TdNPR6-B*	TRIDC5BG023740.4	5B:257840960-257843317(–)	493	51.62	6.11	5
*Ae. tauschii*	I	*AetNPR1*	AET3Gv20232400.2	3D:64296876-64300961(+)	608	66.59	5.41	5
		*AetNPR2*	AET7Gv20038900.2	7D:7720290-7726555(+)	454	50.19	5.46	3
	II	*AetNPR3*	AET3Gv20713100.1	3D:425287443-425291551(+)	607	66.31	5.45	3
		*AetNPR4*	AET4Gv20029500.2	4D:6713853-6718113(–)	596	66.37	5.86	7
	III	*AetNPR5*	AET3Gv21117800.1	3D:592078399-592081502(–)	514	54.17	6.06	13
		*AetNPR6*	AET5Gv20354800.2	5D:229230789-229233557(+)	515	54.18	6.63	8

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
