# Peer review of "Genome-Wide Identification and Analysis of the NPR1-Like Gene Family in Bread Wheat and Its Relatives"

_ijms, 2019, doi:10.3390/ijms20235974_

Round 1

Reviewer 1 Report

Due to the important roles NPR1-like genes have in plant immunity, it is crucial to understand how these genes evolved, including the pattern of gene family expansion and contraction during the evolution of plant species, the sub-/neo-/pseudo-functionalization of gene copies after gene duplication, etc.  This manuscript did the genome-wide identification and analysis of NPR1-like genes in wheat and its relatives, showing the conservation in gene structures and divergence in gene expression profiles, which are useful for the community who are interested in the evolution of plant immunity in wheat and its relatives. This manuscript is well organized and written. However, the nomenclature of genes is not consistent across the manuscript, and for some paragraph, the contextual transition is not good, which lead to some confusion. In addition, the authors didn’t have a comprehensive discussion, especially for evolutionary aspects. I’d recommend this paper to be published in IJMS after the authors revise the following comments.

Major comments:

In Section “Result 2.1”, the authors didn’t specify whether all NPR1-like protein sequences from 8 monocotyledonous and 15 dicotyledonous plants were included in the phylogeny analysis, which I doubt. If yes, how did the authors get all sequences from each species? In line 154-155, the authors listed several but not all genes in each clade, why?  Are these genes listed more important than others?

The term “gene” and the number of genes was not consistent across the manuscript, as sometimes it’s 40 genes, sometimes it’s 24 genes. The authors took homoeologous copies in A, B and D subgenomes as a single gene (line 157), which doesn’t make sense to me. However, in the following part, the gene was referred to as each copy again, e.g., line 174. Please double check and be consistent across the whole manuscript. I’d suggest the authors to use another term for homoeologous copies if the author do want to summary them with a name.

The nomenclature of genes is also not consistent. For example, in line 126, a gene from T.tu was named as wNPR1, while in Fig. 1, genes from T.di also have gene names starting with “w”. Because in this manuscript, there are at least two wheats mentioned, i.e., bread wheat, emmer wheat, or tetraploid and hexaploidy wheats. Sometimes the authors used “wheat genes”, which was not specified which species these genes belong to. This leads to confusion. In addition, copies from subgenomes were referred to as NPRX-A/B/D (e.g. line 165) sometimes, but A/B/D-NPRs (e.g., line 223) elsewhere. The gene names among species are also inconsistent. For example, in branch leading to TaNPR2-A/B, T.di had two genes named TdNPR2-A and TdNPR3-A. So TdNPR2-A and TdNPR3-A are duplicates in A subgenome, right? I’d suggest the authors to use TdNPR2-A and TdNPR2-A’ or TdNPR2-A-1 and TdNPR2-A-2, and for each branch, use the same number after ‘NPR’.

In line 284-293, the discussion about gene duplication in NPR family is misleading and incomplete. Based on Fig.1, Clade I had a duplication event in ancestor of monocots after the split between eudicots and monocots, while each of Clade II and Clade III had a gene duplication event before the split between eudicots and monocots and then copies in eudicots were lost. If this is true, is it consistent with other studies? And what is your hypothesis about the gene loss in eudicots? In addition, in lines 289-290, “the NPR clade might be split into two branches after two rounds of duplications, the NPR1/2 subfamily and the NPR3/4 subfamily”, should be one round of duplication leading to the split of NPR1/2 and NPR3/4 subfamily. Furthermore, the wheat “NPR1/2” and “NPR3/4” is different from Arabidopsis, therefore, be careful of the using of these terms.

In line 312-319, again, the discussion for gene evolution is not complete, some very interesting findings were neglected by the authors. For example, the NPR2-like genes were lost in T. ur and B subgenome in T. ae and T. di. In contrast, in T. di, the NPR2-like gene was duplicated in A subgenome, leading to TdNPR2-A and TdNPR3-A, and then one of these two copies was lost again in T. ae. Had the NPR2-like genes gone through pseudogenization in these three species? Whether the authors can find the residual sequences of NPR2-like genes? In addition, TaNPR2-A/D have very low expression levels. Taken together, these results indicate that NPR2-like genes may have undergone relaxed selection. I’d suggest the authors to calculate the Ka/Ks for each branch to see which branches had experienced negative selection and which had experienced relaxed, or even positive selection.

Minor comments:

There are some typo or English language issues. Please double check the whole manuscript carefully. For example, Line 104,  the wording is problematic. Should be “Arabidopsis NPR1 and its homologs had been shown to be involved in SA-dependent defense responses in many plant species through genetic transformation.” In line 179, “sequence comparison” should be “sequence composition”, right? Line 723-724, should be “and other NPR1-related proteins with experimentally confirmed functions”.

Line 145, what are the bioinformatics tools? BLAST? Please specify.

In Fig. 1, not all genes from T. aestivum were marked. And some genes from T. dicoccoides and Ae. tauschii were marked. Please double check.

In Fig. 1, please add the bootstrap values on the branches, showing the confidence of the phylogeny relationships.

Line 223 to 226. This paragraph is not will written. There is no transition why the authors need to generate phylogenetic trees.

Line 289, when did the ancient gene duplication events happen? Pre-angiosperms?

Line 291, “this ancient duplication”, please specify which duplication event it is, event leading to BOP5/6, or leading to NPR/BOP?

Please cite the references for the expression data.

Reviewer 2 Report

This manuscript study the structure and conservation of NPR1-like genes in T. Aestivum and related species. They study motifs, promoter sequences involved in hormone response and co-linearity of these genes among T aestivum and relatives. They also use existing published data to do a meta-analysis on expression under biotic stress. In general, out of this bio-informatics studies, which are correctly performed, they conclude that TaNPR1 has similar functions to AtNPR1 and TaNPR5 and 6, similar functions to those from Arabidopsis. Although this is an interesting study, they do not support their hypothesis on the role of these NPR1-like genes on the different processes with any own wet lab experiments. With the in silico experiments they have performed, they can only speculate on these functions. Therefore, they should remove these conclusions from the manuscript.

The Discussion section is only a repetition of the results, without a proper discussion of those results. The authors should re-write this section. Besides, they mention figure S6, which does not exist among supplementary figures. I suppose the authors mean figure S5.

Minor comments

Figure 2C lacks resolution, since characters are small and increasing size reveals low resolution.

Line 71, the S of Salicylic is missing.

Line 96: In addition, instead of In additional.

Line 260: lower instead of low.

Round 2

Reviewer 2 Report

No comments

Author Response

Thanks.